# 7,7″-Dimethoxyagastisflavone Inhibits Proinflammatory Cytokine Release and Inflammatory Cell Recruitment through Modulating ERα Signaling

**DOI:** 10.3390/biomedicines9121778

**Published:** 2021-11-26

**Authors:** Yi-Shin Wu, Chian-Ruei Chen, Yun-Ting Yeh, Han-Huei Lin, Yin-Hsuan Peng, Yu-Ling Lin

**Affiliations:** Agricultural Biotechnology Research Center, Academia Sinica, 128 Academia Road, Section 2, Nankang, Taipei 11529, Taiwan; hot0281@gate.sinica.edu.tw (Y.-S.W.); chen.sherry10@gmail.com (C.-R.C.); d10731524@gapps.fg.tp.edu.tw (Y.-T.Y.); henry025@gate.sinica.edu.tw (H.-H.L.); yinhsuan.peng@uqconnect.edu.au (Y.-H.P.)

**Keywords:** 7,7″-dimethoxyagastisflavone, biflavonoid, inflammation, proinflammatory cytokine, migration, estrogen receptor α

## Abstract

Acute systemic inflammatory diseases, including sepsis, usually result in cytokine disorder and multiple-organ failure. 7,7″-Dimethoxyagastisflavone (DMGF), a biflavonoid isolated from the needles of *Taxus x media var. Hicksii*, has previously been evaluated for its antiproliferative and antineoplastic effects in cancer cells. In this study, the effects of DMGF on the cytokine production and cell migration of inflammatory macrophages were investigated. The inhibition of cytokine and chemokine production by DMGF in LPS-treated macrophages was analyzed by a multiplex cytokine assay. Then, the integrin molecules used for cell adhesion and regulators of actin polymerization were observed by RT-PCR and recorded using confocal imaging. The DMGF interaction with estrogen receptor α (ERα) was modeled structurally by molecular docking and validated by an ERα reporter assay. DMGF inhibited TNF-α, IL-1β, and IL-6 production in LPS-induced macrophages. DMGF also inhibited inflammatory macrophage migration by downregulating the gene and protein expression of adhesion molecules (LFA-1 and VLA4) and regulators of actin assembly (Cdc42-Rac1 pathway). DMGF might interact with the ligand-binding domain of ERα and downregulate its transcriptional activity. These results indicated that DMGF effectively inhibited the production of proinflammatory cytokines and the recruitment of inflammatory cells through downregulating ERα signaling.

## 1. Introduction

7,7″-Dimethoxyagastisflavone (DMGF) is a biflavonoid isolated from the needles of *Taxus media var. Hicksii* [1]. The biological activity of DMGF was first revealed when it was found to be an inhibitor of the production of aflatoxin by *Aspergillus flavus* [2]. Our research further demonstrated that DMGF could induce apoptotic and autophagic cell death in various types of cancer cells [1]. DMGF significantly suppressed the metastatic behaviors of highly invasive melanoma cancer cells, including the intravasation, colonization, and invasion of the lymphatic duct [3]. DMGF can downregulate the levels of key modulators of the Cdc42/Rac1 pathway to interfere in F-actin polymerization and suppress the formation of lamellipodia by reducing the phosphorylation of CREB [3]. These findings suggest that DMGF may be further developed as a chemotherapeutic drug for patients with metastatic melanoma; however, the anti-inflammatory activity of DMGF has hitherto not been investigated.

Inflammation is a protective immune response to harmful stimuli, such as pathogens or viruses, driven by the evolutionarily conserved innate immune system. However, an acute systemic inflammatory response such as sepsis usually results in a dysregulated cytokine storm which causes a massive inflammatory cascade and irreversible organ dysfunction [4]. The clinical treatment of sepsis focuses on treating the primary triggering condition to disrupt the progression of the continuum of septic shock and multi-organ dysfunction syndrome [4]. After ensuring hemodynamic stability, broad-spectrum antibiotics and glucocorticoids are used to improve survival and the reversal of shock in patients [4,5]. Glucocorticoid treatment can counteract several sepsis characteristics, including excessive inflammation, vascular defects, and hypoglycemia [6]. However, glucocorticoid resistance can cause hypothalamic–pituitary–adrenal axis dysfunction, which hampers the maintenance of homeostasis and regulation of cytokines [6]. The management of systemic inflammatory responses is expected to improve the treatment of sepsis in the future.

During the inflammatory response, proinflammatory cytokines and chemokines increase immune cell recruitment. The recruitment of immune cells to the infection site requires their trafficking across the blood vessel wall [7]. Under inflammatory conditions, inflammatory monocytes/macrophages transmigrate following a series of events, including rolling adhesion, tight binding, diapedesis (the squeezing of cells through blood cell walls), and migration [7]. Rolling adhesion on vascular surfaces is the first step in recruiting circulating leukocytes, hematopoietic progenitors, or platelets to specific organs or to sites of infection or injury. Rolling requires the rapid yet balanced formation and dissociation of adhesive bonds in the challenging environment of blood flow. Initially, proinflammatory cytokines enhance the expression of adhesion molecules, such as P-selectin, E-selectin, intercellular adhesion molecule-1 (ICAM-1), and vascular cell adhesion molecule-1 (VCAM-1) on the surface of the endothelial cells [7]. These adhesion molecules expressed on endothelial cells are required for cell migration. Inflammatory monocytes express a high level of P-selectin glycoprotein ligand-1 (PSGL-1), which transiently interacts with P- and E-selectin and mediates rolling on the vascular endothelium [8]. The increased expression of PSGL-1 has been found to be associated with inflammation [9]. Monocyte/macrophage rolling strongly depends on monocyte-expressed very late antigen-4 (VLA-4 α4β1 integrin), which binds to VCAM-1 and allows slow rolling on the cytokine-activated endothelium, thereby promoting the transition between rolling and tight binding. Monocyte rolling and arrest also depend on the leucocyte integrin lymphocyte function-associated antigen-1 (LFA-1) and macrophage-1-antigen (Mac-1), which bind to endothelial ICAM-1 and ICAM-2. When exposed to C-C and C-X-C chemokines, integrins expressed on the surfaces of monocytes tightly bind to adhesion molecules on the endothelium. The proinflammatory mediators have been reported to prime leukocytes for integrin expression and activation [9]. Increased LFA-1 levels on monocytes are associated with virus infection [10]. Finally, monocytes squeeze between the endothelial cells (diapedesis), penetrate the connective tissue, and move to the infection site. In the diapedesis stage, the distribution of adhesion molecules and actin cytoskeleton of monocytes shows dynamic changes [11]. Therefore, when macrophages receive inflammatory stimuli, they release cytokines and chemokines to promote the recruitment of more inflammatory cells. The recruitment of these cells to the inflammation site requires their movement through rolling and adhesion in the blood vessels. Based on these factors, the effective inhibition of cytokine and chemokine release and macrophage migration are the important factors that inhibit the inflammatory response.

In this study, the anti-inflammatory effects of DMGF were investigated. The in vitro results indicated that DMGF efficiently decreased the release of LPS-induced proinflammatory cytokines and other chemo-mediators. DMGF also inhibited inflammatory macrophage migration through downregulating the expression of adhesion molecules and the assembly of F-actin. In a mouse model of endotoxin shock, DMGF not only decreased the expression of proinflammatory cytokines in sera but also decreased the infiltration of macrophages. DMGF reversed LPS-induced tissue damage and maintained the functions of various organs. Furthermore, DMGF downregulated the transcriptional activity of NF-κB and estrogen receptor alpha (ERα), thus reducing cytokine release and transmigration in inflammatory macrophages.

## 2. Materials and Methods

### 2.1. DMGF Preparation

DMGF (98% purity by high-performance liquid chromatography) was isolated from the dried needles of *Taxus media var. Hicksii* and purified according to the method described previously [1,3].

### 2.2. Cells and Cell Cultures

RAW264.7 mouse macrophages were purchased from the Bioresource Collection and Research Center (BCRC, Hsinchu, Taiwan, ROC) and maintained in Dulbecco’s modified Eagle’s medium (DMEM, Gibco/Invitrogen, Carlsbad, CA, USA) supplemented with heat-inactivated 10% fetal bovine serum (Gibco/Invitrogen) and 1% Penicillin/streptomycin (Gibco/Invitrogen) in 5% CO_2_ at 37 °C. Mouse splenocytes were isolated from BALB/c mice according to the following procedures. Briefly, BALB/c mice were sacrificed and their spleens were collected. The spleen was placed into the cell strainer and homogenized through the cell strainer into the Petri dish. Erythrocytes in the homogenized spleen were lysed by ACK lysis buffer at room temperature for 5 min following the PBS wash. After the spinning down of the cells, they were resuspended and cultured with RPMI1640 growth medium (Gibco/Invitrogen) supplemented with heat-inactivated 10% fetal bovine serum (Gibco/Invitrogen) and 1% Penicillin/streptomycin (Gibco/Invitrogen).

### 2.3. Cell Proliferation Assay

Cells were seeded at 1 × 10^4^ cells/well overnight in a 96-well microculture plate. After treatment with serial concentrations of DMGF in dimethyl sulfoxide (DMSO, final concentration of DMSO is 0.1%) for 48 h, their cell proliferation was measured by CCK-8 (Dojindo Molecular Technologies, Kumamoto, Japan) cell counting regents. The cell viability ratio (%) was calculated using the following equation: % viability = absorbance of test sample/absorbance of control × 100%. The results were expressed in duplicate with three independent experiments.

### 2.4. Proinflammatory Cytokines

Splenocytes were seeded at 1 × 10^6^ cells/well in a 24-well culture plate and then treated with 1 μg/mL LPS and various doses of DMGF for 48 h. The levels of TNF-α, IL-1β, and IL-6 in the culture media were determined by the ELISA kit (R&D System, Minneapolis, MN, USA) according to the manufacturer’s instructions.

### 2.5. Cytokine Profile Analysis

Splenocytes were seeded at 1 × 10^6^ cells/well in a 24-well culture plate and then treated with 1 μg/mL LPS and 0.6 μg/mL DMGF for 48 h. The supernatants in each group were collected for analysis of cytokine levels by a cytokine multiplex assay using the Bio-Plex Pro Mouse Cytokine Standard Group I 23-Plex, following the manufacturer’s protocol (Bio-Rad, Hercules, CA, USA).

### 2.6. Macrophage Migration Assay

Macrophage migration was performed by the Boyden chamber technology. Briefly, RAW264.7 cells were treated with 1 μg/mL LPS and/or 0.6 μg/mL DMGF and then plated (20,000 cells/well) in the upper well of a transwell plate (5-μm pore; Merck Millipore, Billerica, MA, USA). The cells were allowed to migrate toward the lower well with DMEM growth medium for 10 h. The cells on the top of the filter were removed with a cotton swab and migrated cells on the underside were fixed with methanol, stained with 50 μg/mL propidium iodide and counted with a fluorescence microscope.

### 2.7. Reverse Transcription-Polymerase Chain Reaction (RT-PCR)

RAW264.7 cells (3 × 10^5^ cells/well) were treated with 1 μg/mL LPS and 0.6 μg/mL DMGF for 10 h. Then, total cellular RNA was extracted with TRIzol (Invitrogen Life Technologies, Carlsbad, CA, USA) and reverse-transcribed into cDNA using the Superscript-III kit (Invitrogen). PCR analysis was performed on aliquots of the cDNA preparations to detect gene expression. The cDNA of *Itgal*, *Itga4*, *Itgb1*, *Itgb2*, *Cdc42*, *Pak1*, *n-Wasp*, *Rac1*, *Limk*, and *Esr1* were then amplified by PCR. The primers for mouse *Itgal* were: forward primer—5′-CATGGGCCACGTTCTCATCT-3′ and reverse primer—5′-GCCAGTCAGCCTACATGGTT-3′; the primers for mouse *Itga4* were: forward primer—5′-CAGCAAAAAGGCATAGCGGG-3′ and reverse primer—5′-AGC TCTGTGAGCCATCGAAC-3′; the primers for mouse *Itgb1* were: forward primer—5′-ATTGCCAGACGGAGTAACAA-3′ and reverse primer—5′-CTGTGCTACATTCACAGTGC -3′; the primers for mouse *Itgb2* were: forward primer—5′-CGGTTTCTTTCCGCCATTAT-3′ and reverse primer—5′-CAAAAGACTCCACAAAGCTTAC-3′; the primers for mouse *Cdc42* were: forward primer—5′-CGGAGAAGCTGAGGACAAGAT-3′ and reverse primer—5′-GAGTGTATGGCT CTCCACCAA-3′; the primers for mouse *Pak1* were: forward primer—5′-ACTATG ATTGGAGCCGGCAG-3′ and reverse primer—5′-TGGCATTCCCGTAAACTCCC-3′; the primers for mouse *n-Wasp* were: forward primer—5′-AATGCTGGACACTGG CTACC-3′ and reverse primer—5′-ACGGGGTGGGAGTGAGATAA-3′; the primers for mouse *Rac1* were: forward primer—5′- AGATGCAGGCCATCAAGTGT-3′ and reverse primer—5′-TAGGAGAGGGGACGCAATCT-3′; the primers for mouse *Limk* were: forward primer—5′-CCATCAAGGTGACACACCGA-3′ and reverse primer—5′-GCAAAGCTGACCCTCTGACT-3′; the primers for mouse *Esr1* were: forward primer—5′-GCTGAACCGCCCATGATCTA-3′ and reverse primer—5′-CCAGGAGCAAGTTAGGAGCA -3′. All PCR reagents used to amplify the cDNA were purchased from Promega (Madison, WI, USA). Actin cDNA in the samples was used to normalize the loading amounts in each reaction. Finally, PCR products were resolved by electrophoresis on 2% agarose gels, stained with ethidium bromide and photographed using the BioDoc-It Imaging System (UVP, Upland, CA, USA).

### 2.8. Confocal Immunofluorescence

To observe the expression of LFA-1 and VLA-4, RAW264.7 cells plated on the cover slides were incubated at 37 °C overnight and then treated with 1 μg/mL LPS and 0.6 μg/mL DMGF for 12 h. Cells were fixed with 4% formaldehyde for 10 min and stained by Concanavalin A (Con A) conjugated with Alexa Fluor 488 for cell membrane staining. Then, the cells were, respectively, stained with APC conjugated anti-mouse LFA-1 (Biolegend, San Diego, CA, USA) and Alexa Fluor 647 conjugated anti-mouse VLA-4 (Biolegend) antibodies for 60 min. To observe the structures of lamellipodia, the fixed cells were stained with phallotoxin for 60 min for F-actin staining. The stained cells were observed with DAPI (Vectashie mounting medium, Vector Laboratories, Burlingame, CA, USA) by a Zeiss LSM 780 Confocal microscope (Zeiss, Jena, Germany).

### 2.9. Western Blot

Cells were treated with 1 μg/mL LPS and 0.6 μg/mL DMGF for 10 h and then lysed in RIPA lysis buffer. The protein concentration of the cell lysate was estimated with the Bradford protein assay using BSA as the standard. Total proteins (50 μg) were separated by SDS-PAGE using a 10% polyacrylamide gel and then transferred onto a nitrocellulose membrane. The membrane was blocked with 5% skim milk in phosphate-buffered saline with Tween 20 (0.05% *v/v* Tween-20 in PBS, pH 7.2) for 1 h. The membranes were then incubated with primary antibodies such as anti-mouse NF-κB (1:1000, Cell Signaling), anti-mouse IκB (1:1000, Cell Signaling) and anti-mouse ERα antibodies (1:1000, Bioss) at 4 °C overnight, followed by incubation with a horseradish peroxide-linked secondary antibody (1:10,000). The protein bands were visualized using the Amersham ECL Western Blotting Detection Reagents (GE Healthcare, Buckinghamshire, UK). The intensity of the chemiluminescence signal was quantified using the Biospectrum 815 Imaging System and Vision Works Software (UVP, Upland, CA, USA).

### 2.10. NF-κB-GFP Transcriptional Reporter Assay

Due to the low transfection efficiency in RAW264.7 cells (approximately 20%), HEK293 cells were used as the experimental cell model for the NF-κB transcriptional reporter assay. HEK293 cells were seeded at 3 × 10^5^ cells per well and cultured overnight in six-well cultured plates. In transfection assays, 200 μL/well of the transfection sample containing 3 μg plasmid DNA containing NF-κB response element and GFP reporter and 10 μL Lipofectamine (Invitrogen, NY, USA) in Opti-MEM (Invitrogen) were added and incubated with the cells. The 2 mL/well serum containing DMEM complete media was added after 16 h, and the transfected cells were resuspended and divided equally into each well of a 24-well plate. Then, 50 ng/mL of phorbol 12-myristate 13-acetate (PMA) (Sigma-Aldrich, St. Louis, MO, USA) was added for NF-κB induction and 0.6 μg/mL DMGF was administered for 24 h NF-κB–GFP reporter gene expression was assayed by flow cytometry (BD Biosciences, Franklin Lakes, NJ, USA).

### 2.11. Analysis of Cell Surface ER-α Expression by Flow Cytometry

RAW264.7 cells were seeded at 1 × 10^5^ cells/well in a 24-well culture plate overnight and then cells were treated with 1 μg/mL LPS and/or 0.6 μg/mL DMGF. To evaluate ER-α expression on the membrane, cells were harvested, fixed and incubated with anti-mouse ER-α antibody (1:200, GeneTex, Hsinchu, Taiwan, ROC) for 1 h and then the FITC-conjugated secondary antibodies (1:200, Jackson ImmunoResearch, West Grove, PA, USA) for 30 min at 4 °C. Fluorescence related to immunolabeling was analyzed by flow cytometry (BD Biosciences).

### 2.12. ER-α Transcriptional Reporter Assay

HEK293 cells were seeded at 3 × 10^4^ cells/well in a 96-well culture plate overnight and then co-transfected with pBIND-ER-α and pGL4.35 ERE-Firefly luciferase reporter plasmid (Promega). The vector of pGL4 reporter lacking ERE served as a negative control vector. The transfection efficiency was served by pBIND-ER-α containing the gene of the hRluc-neomycin fusion protein. Cells were harvested in reporter lysis buffer 24 h post-transfection, and lysates were assayed for luciferase activity with a dual luciferase assay kit (Promega, Madison, WI, USA) according to the manufacturer’s instructions. Luciferase activity was normalized with the ratio of firefly to Renilla luciferase activities.

### 2.13. Drug Docking

The protein target candidates for DMGF docking were predicted by the SwissTargetPrediction Website (http://www.swisstargetprediction.ch/, accessed on 24 June 2019). DMGF (red) was docked with estrogen receptor α (ER-α) by iGEMDOCK v2.1 software (http://gemdock.life.nctu.edu.tw/dock/igemdock.php). ER-α (PBD code: 1QKU) was selected from the Protein Data Bank (PDB). The parameters of iGEMDOCK were set as follows: population—500; generation—70; and solution—100.

### 2.14. ESR1 Knockdown

Lentiviral particles with the lentiviral vector expressing *Esr1* siRNA and scrambled siRNA were purchased from the RNA Technology Platform and Gene Manipulation Core (Academia Sinica, Taipei, Taiwan). The target sequence of *Esr1* is 5′- GCCGAAATGAAATGGGTGCTT-3′ (siESR1). The scrambled siRNA sequence is 5′-CCTAAGGTTAAGTCGCCCTCG-3. RAW264.7 cells were subcultured at 5 × 10^5^ cells/well into six-well tissue culture plates overnight. Cells were infected with lentiviral particles at a multiplicity of infection (MOI) of 10. To detect the interference effects of different target, ESR1 mRNA expression was determined using RT-PCR.

### 2.15. Animals

Female BALB/c mice, 5 weeks of age, were purchased from the National Laboratory Animal Center (Taipei, Taiwan). All studies were performed in the Laboratory Animal Core Facility of Academia Sinica following the Guide for the Care and Use of Laboratory Animals and were approved by the Institutional Animal Care and Use Committee (IACUC) of Academia Sinica (19-03-1294).

### 2.16. Endotoxic Mouse Model

Balb/c mice (5-week-old females) were randomized to receive either 10 mg/kg of DMGF or a saline/DMSO solvent by intraperitoneal (i.p.) injection. Immediately afterward, the mice were i.p. injected with 5 mg/kg of LPS. After treatment for 2, 4, 6 and 8 h, blood samples were harvested from submandibular bleeding and centrifuged at 3000× *g* for 5 min to obtain the sera. The murine proinflammatory cytokines, such as TNF-α, IL-1β, and IL-6 were measured according the instructions included with the ELISA kits (R&D System, Minneapolis, MN, USA). 

### 2.17. Tissue Damage and Macrophage Infiltration

To investigate the effects of DMGF on suppressing inflammation-induced organ damage, tissue morphology was observed. After sacrificing the DMGF-treated mice, the organs, including the heart, lungs, liver, spleen, and kidneys, were removed and fixed in 10% buffered formalin−saline overnight and then embedded in paraffin blocks. Tissue sections were prepared and stained with hematoxylin and eosin (H&E). The morphology was observed under the Zeiss Axiovert 200M microscope (Zeiss, Oberkochen, Germany).

The infiltrating macrophages in organs were analyzed by immunohistochemistry staining. Tissue sections were stained with anti-F4/80 (1:1000) antibodies for macrophage detection.

### 2.18. Statistical Analysis

The results were expressed as mean ± SD and analyzed using the SAS statistical software package (SAS Institute, Cary, NC, USA). The t-test was used when comparing two independent samples and the ANOVA test was used when comparing multiple samples. Differences with a *p* value of less than 0.05 were considered statistically significant.

## 3. Results

### 3.1. Anti-Inflammatory Effects of DMGF in Macrophages

Mouse splenocytes were simultaneously stimulated with LPS and treated with DMGF, and their cell proliferation was assessed. As shown in Figure 1A, DMGF below 1.2 μg/mL did not affect cell proliferation, regardless of whether cells were treated with LPS. To understand the anti-inflammatory role of DMGF, non-cytotoxic concentrations of DMGF were used to treat LPS-induced splenocytes, and the concentrations of different proinflammatory cytokines were measured. The results in Figure 1B show that DMGF significantly inhibited the release of TNF-α, IL-1β and IL-6 in a dose-dependent manner. In addition to pro-inflammatory cytokines, DMGF also inhibited the release of chemokines such as KC (CXCL-1), MIP-1α (CCL-3) and MIP-1β (CCL-4) (Table 1).

### 3.2. DMGF Decreases Macrophage Migration

The results of the transwell assay showed that DMGF was able to decrease macrophage migration (Figure 2A). To investigate the effects of DMGF on cell migration, the expression of adhesion molecules on the cell surface and F-actin polymerization were determined. VLA-4 and LFA-1, adhesion molecules that are important for leukocyte rolling in the blood stream, were evaluated upon DMGF treatment. DMGF significantly suppressed the mRNA expression of *Itgal* and *Itgb2*, two subunits of heterodimeric integral membrane protein that form LFA-1 (Figure 2B). DMGF also reduced the mRNA expression of *Itga4* and *Itgb1*, two subunits of heterodimeric membrane protein that form VLA-4. However, there was no significant difference in *Itgb1* expression compared to the untreated group (Figure 2B). The image results showed that the expressions of LFA-1 and VLA-4 were increased in LPS-stimulated macrophages (Figure 2C,D). After DMGF treatment, the expression of both LFA-1 and VLA-4 was downregulated (Figure 2C,D). In addition to adhesion molecules, DMGF decreased filopodia and lamellipodia production and led to a round cell morphology (Figure 2C,D). These results suggest that the inhibition of macrophage migration in response to DMGF depends on a mechanism relying on F-actin polymerization. We further determined the expression of proteins that drive the filopodia (Cdc42, Pak1, and n-Wasp) and the lamellipodia (Rac1 and Limk) after DMGF treatment. Figure 3A shows that DMGF efficiently downregulated the mRNA expression of *Cdc42*, *Pak1* and *Limk*. Fluorescent phallotoxin staining was used to show actin fibers in LPS-stimulated macrophages (Figure 3B). The fragmentation of F-actin caused by DMGF treatments is visible in Figure 3B. Taken together, these results indicate that DMGF suppressed macrophage movement through inhibiting the expression of adhesion molecules and F-actin polymerization.

### 3.3. Anti-Inflammatory Activity of DMGF through the Inhibition of NF-κB and ERα Transcriptional Activity

NF-κB is an essential transcriptional factor for inflammation. DMGF dramatically downregulated LPS-induced NF-κB expression and upregulated IκB expression (Figure 4A). Additionally, the results of the NF-κB-GFP transcriptional reporter assay also showed that DMGF suppressed NF-κB transcriptional activity (Figure 4B).

According to the results of Swiss Target Prediction, estrogen receptor α (ERα) was predicted as a possible target of DMGF. DMGF was docked on a ligand-binding domain (LBD) of ERα by iGEMDOCK software and interacted with Thr347 (Figure 5A). ERα is a transcription factor which regulates the transcription of genes involved with proinflammatory cytokines, chemokines, integrin and actin polymerization molecules (Appendix A). The murine macrophages expressed the classical intracellular ERα. DMGF did not affect the protein expression of ERα on the cell membrane or total cellular levels of ERα in LPS-stimulated macrophages (Figure 5B,C). To determine whether DMGF directly affected ER transcriptional activity, we co-transfected an ERα cDNA plasmid containing a ligand-binding domain (LBD) and a pGL4.35 ERE-Firefly luciferase reporter plasmid into HEK293 cells. Relative to the control cells, 17β-estradiol (E2) activated ER for reporter gene expression. Fulvestrant, an ER antagonist, was used as the positive control. DMGF treatment was able to efficiently suppress E2-induced ER transcriptional activity (Figure 5D). When RAW264.7 cells with *Esr1* knockdown were treated with DMGF, the inhibitory effects of DMGF on α-α release, cell transmigration, integrin expression and *Limk* expression were reversed (Figure 5E–G). These results therefore show that DMGF significantly inhibited LPS-induced inflammation through the modulation of NF-κB and ERα signaling.

### 3.4. DMGF Inhibited the Release of Pro-Inflammatory Cytokines in an Endotoxin Mouse Model

Next, an endotoxic shock mouse model was used to examine the therapeutic effects of DMGF on acute inflammation in vivo. After induction with LPS, mice had the highest level of TNFα at 2 h. Compared to the LPS-treated mice, DMGF significantly reduced the secretion of LPS-induced TNF-α at 2, 4, and 6 h (Figure 6A). DMGF also effectively inhibited the release of LPS-induced IL-1β and IL-6 after treatment (Figure 6A).

Hyperinflammation causes multiple-organ damage and dysfunction. In the endotoxin shock model, multiple-organ damage was seen, as shown in Figure 6B. Mice in the LPS group had bleeding in all organs. There were vacuoles and edema in the liver and kidney of LPS-treated mice. Abnormal cell proliferation was observed in the lungs and intestine. It is worth noting that immune cells infiltrated into the liver, lung, kidney, and intestine. This organ damage, such as bleeding, edema and immune cell infiltration, was limited by treatment with DMGF (Figure 6B). To observe the macrophage infiltration after stimulation with LPS, an IHC analysis of liver, lung and kidney tissues was conducted. LPS stimulation increased the macrophage infiltration in three organs, particularly in the lung (Figure 6C). DMGF effectively reduced macrophage infiltration and decreased the damage in the liver, lung, and kidney.

## 4. Discussion

DMGF, a bioactive biflavonoid, has anti-tumor and anti-metastatic activities in melanoma cells [3]. However, little is known about its other functions such as in the inhibition of inflammation except for its effect on inducing tumor death through apoptosis or autophagy in cancer cells [1]. In vitro and in vivo studies showed that DMGF effectively inhibited inflammation and protected the endotoxin shock mice from hyperinflammation and organ damage caused by inflammatory macrophage infiltration (Figure 1A and Figure 6). DMGF significantly decreased the release of pro-inflammatory cytokines and chemokines (Table 1 and Figure 1B) and reduced the motility of LPS-stimulated macrophages through suppressing adhesion molecules and disrupting actin polymerization (Figure 2 and Figure 3). Furthermore, DMGF interacted with the LBD of ERα and downregulated ERα transcriptional activity (Figure 5A,C). This study suggests the potential of DMGF in the suppression of inflammation through inhibiting the release of pro-inflammatory cytokines and chemokines and interrupting the recruitment of inflammatory cells.

It has been reported that ERα regulates cell signaling pathways in innate and adaptive immunity. ERα is expressed in monocytes/macrophages. In vivo estrogen stimulation promotes the production of proinflammatory cytokines in TLR4 signaling activation of macrophages [12,13,14], while in vitro experiments have proved that estrogen exerts anti-inflammatory properties on monocyte/macrophage cells [15]. Recently, selective ER modulator drugs such as tamoxifen, raloxifene, and some synthetic compounds have been found to reduce inflammation in systemic or tissue-specific inflammatory diseases [16,17]. DMGF interacted with the ERα-LBD at Thr347 and repressed the ERα transcriptional activity (Figure 5A,C) to decrease proinflammatory cytokines and chemokines (Table 1). Thr347 is a key residue for agonists and antagonists binding in the LBD of ERα. The crystal structure-based and computer-aided studies of human ERα show that the various agonists and antagonists, as well as E2 bound to the ERα LBP via hydrogen bonds with Thr347, provide the high ligand-binding affinity [18,19]. In vitro and in vivo results consistently showed that DMGF treatment has an anti-inflammatory effect. DMGF might act as a selective ER modulator for inhibition of LPS-induced inflammation. DMGF not only suppressed the production of inflammatory cytokines but also inhibited cell migration in LPS-stimulated macrophages (Figure 2A). DMGF effectively inhibited the expression of integrins for cell adhesion and molecules for F-actin polymerization/depolymerization through inhibiting ERα transcription. These regulators have ESR1 transcription factor binding motif on their promoter (Appendix A). DMGF impeded ERα transcriptional activity and provided anti-inflammatory and anti-migratory activity in macrophages.

In addition to ERα, NF-κB is a critical mediator of pro-inflammatory gene induction and regulates innate and adaptive immunity. The canonical NF-κB pathway is responsible for the transcriptional induction of inflammatory genes including TNF-α, IL-1β, and IL-6 [20]. DMGF effectively inhibited the NF-κB pathway and transcriptional activity to suppress inflammation (Figure 4A,B). NF-κB signaling can be affected by ERα mediation [21,22]. Estrogen inhibits the LPS-induced pro-inflammatory response, resulting in the inhibition of NF-κB transcriptional activity [15]. Estrogen-bound ERα has been reported to inhibit NF-κB activity through increasing the expression of IκB and decreasing the phosphorylation of IκB [22]. Furthermore, ER can directly bond to NF-κB and interact with a co-activator to inhibit the transactivation. In this study, DMGF treatment increased IκB expression (Figure 4A) and inhibited NF-κB transactivation (Figure 4B). In the virtual analysis, DMGF directly bound to the LBD of ERα (Figure 5A). These results indicate that DMGF has a similar effect to estrogen or selective ER modulators in terms of blocking the NF-κB transcription. Such inhibitory effects of DMGF on crosstalk between NF-κB and ERα might also be the reason for DMGF’s anti-inflammatory effect.

DMGF is a biflavonoid that belongs to the polyphenol family that is known to have anti-inflammatory and immunomodulatory activity. Recently, a few natural biflavonoids have been discovered from natural plants. A well-known biflavonoid amentoflavone has anti-inflammatory and antioxidative functions [23]. Amentoflavone inhibits phytohaemagglutinin (PHA) or the LPS-induced production of NO and ROS and release of proinflammatory cytokines including IL-1β, IL-6, TNF-α, and PGE2 [23]. Amentoflavone exhibits anti-inflammatory activity through the modulation of the ERK signaling pathway [24]. Theaflavin and its derivatives from black tea block NO synthesis by inhibiting NF-κB activity, thereby reducing LPS-induced inflammation [25]. Another biflavonoid composed of two molecular kaempferols isolated from the shell of *Camellia oleifera* shows strong free radical scavenging activity in vivo and has an analgesic effect on inflammation-induced pain [26]. DMGF impacts pro-inflammatory cytokine release (Table 1), the expression of adhesion molecules (Figure 2) and the regulation of cytoskeleton filaments (Figure 3) through downregulating NF-κB and ERα transcriptional activity (Figure 4 and Figure 5). DMGF effectively prevents the cytokine disorder, inflammatory cell infiltration, and tissue damage caused by LPS-stimulated acute inflammation in endotoxin shock mice (Figure 6). Therefore, DMGF modulates the process of inflammation through the reduction in the ERα signaling pathway.

The inflammatory effects of several selective ER modulators, such as fulvestrant, tamoxifen, and raloxifene, have been proved [27,28,29]. In our results, fulvestrant inhibited NF-κB expression in the LPS-induced splenocytes (Appendix A). However, fulvestrant did not significantly inhibit the production of proinflammatory cytokines, such as TNF -α and IL-6 (Appendix A). Our cytokine results were similar to the results in LPS-stimulated microglia. Fulvestrant significantly decreased NO release but did not affect IL-6 release in LPS-stimulated microglia. Even though tamoxifen and raloxifene can decrease the production of NO and proinflammatory cytokines, their actions on cell mobility are not mentioned [27]. In our results, DMGF alleviated LPS-induced inflammatory responses and tissue damages in endotoxic mice. The infiltration of macrophages in organs is also greatly reduced by DMGF. It is attributed to the ability of DMGF to inhibit LPS-induced inflammation and cell migration.

## 5. Conclusions

This study showed that DMGF has anti-inflammatory effects by inhibiting the activity of ERα and NF-κB (Figure 7). DMGF simultaneously impeded pro-inflammatory cytokine release and inflammatory cell movement and limited the tissue damage caused by hyperinflammation. These findings indicate that DMGF has the potential for development for use as a drug for acute inflammation and inflammatory disease in the future.

## Figures and Tables

**Figure 1 biomedicines-09-01778-f001:**
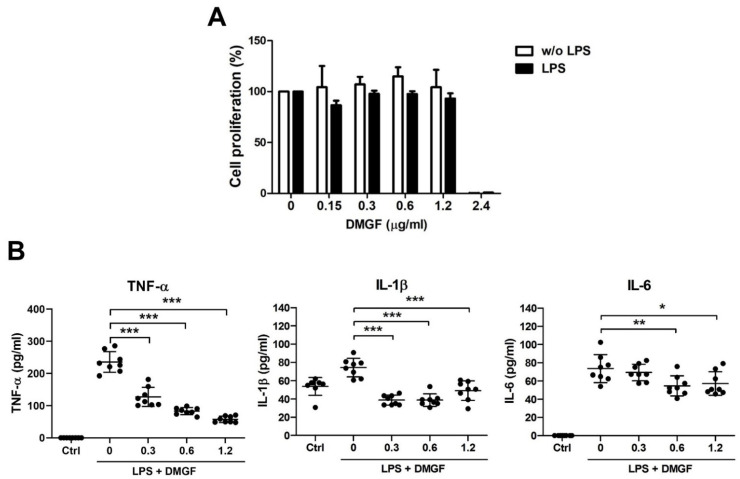
DMGF inhibited LPS-induced inflammation. (**A**) Effects of DMFG on cell proliferation in mouse splenocytes. Splenocytes were treated with serial diluted DMGF with or without LPS for 48 h. Cell proliferation was analyzed by CCK-8 reagent. (**B**) DMGF inhibited production of α-α, IL-1β and IL-6 in LPS-induced splenocytes. LPS-induced splenocytes were treated with 0 to 1.2 μg/mL DMGF for 48 h. *—*p* < 0.05; **—*p* < 0.01; ***—*p* < 0.001, compared to the LPS group.

**Figure 2 biomedicines-09-01778-f002:**
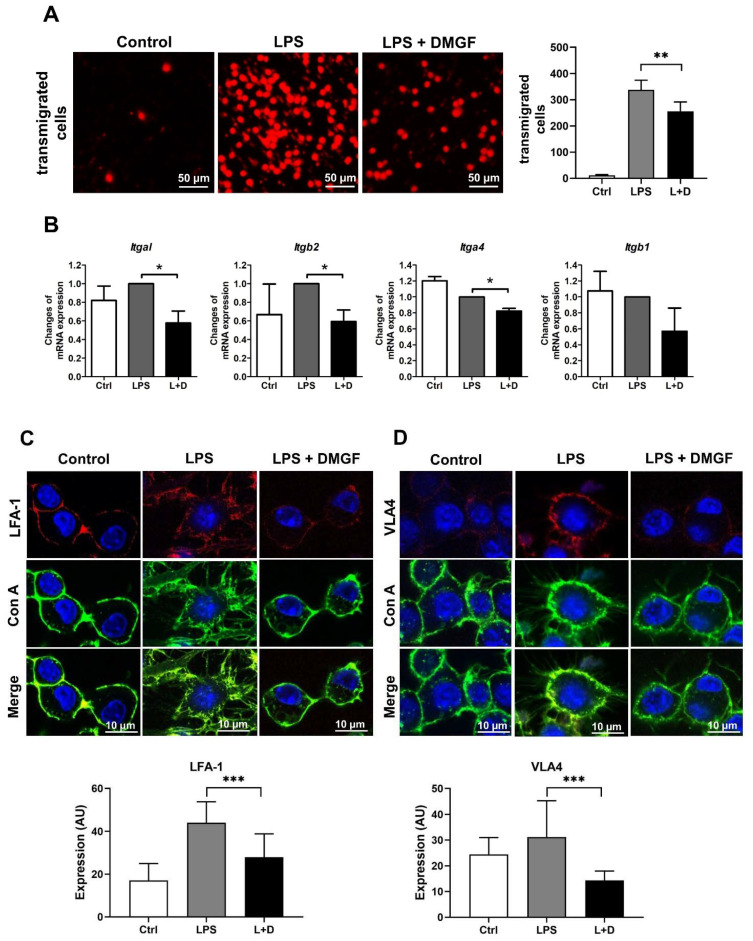
DMGF inhibited LPS-induced migration of inflammatory cells. (**A**) DMGF inhibited migration of LPS-stimulated macrophages. Transmigrated cells were fixed, stained and imaged by an inverted light microscope at 100× magnification. The scale bar in each picture represents 50 μm. (**B**) DMGF inhibited the gene expression of integrins related to adhesion molecules. All values were expressed as the mean ± SD. *—*p* < 0.05; **—*p* < 0.01; ***—*p* < 0.001, compared to the LPS group. (**C**,**D**) DMGF suppressed the expression of LFA-1 and VLA4 on the cell surface. LFA-1 or VLA 4 expression was stained with fluorescent dye-conjugated antibody, shown in red. The cell membrane was stained with Con A conjugated with Alexa Fluor 488, shown in green. The nucleus was stained with DAPI, shown in blue. Con A—Concanavalin A; Ctrl—Control. The scale bar in each picture represents 10 μm.

**Figure 3 biomedicines-09-01778-f003:**
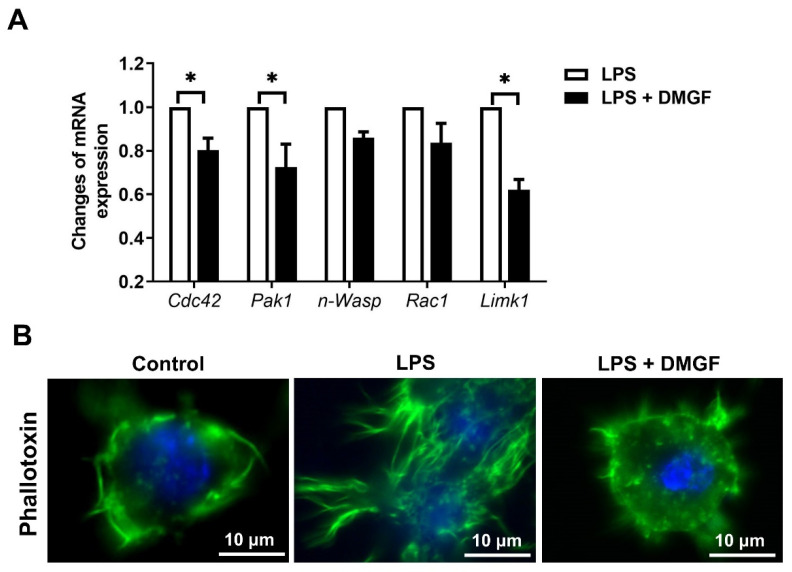
DMGF inhibited actin assembly through Cdc42-PAK-LIMK regulation. (**A**) Effects of DMGF on gene expression of molecules involved in F-actin polymerization and depolymerization. Gene expression levels of *Cdc42*, *Pak1*, *N-Wasp*, *Rac1*, and *Limk* were measured by RT-PCR. *—*p* < 0.05, compared to the LPS group. (**B**) DMGF repressed actin assembly. F-actin was detected by staining with phallotoxin, shown in green. The nucleus was stained with DAPI, shown in blue. The scale bar in each picture represents 10 μm.

**Figure 4 biomedicines-09-01778-f004:**
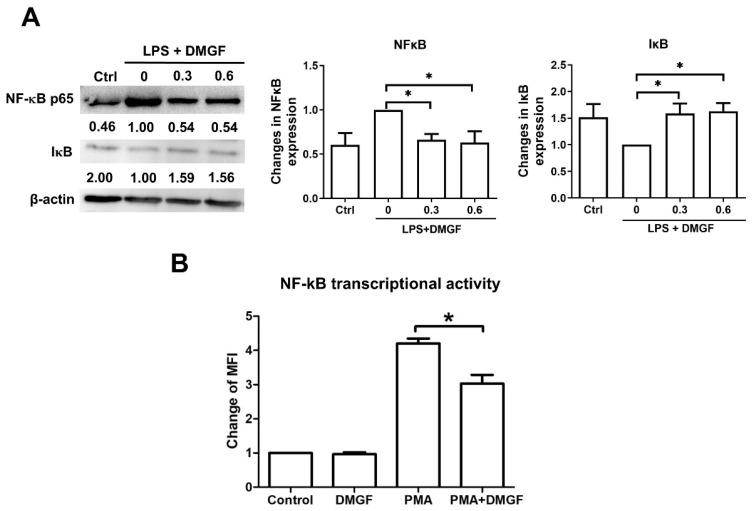
Effects of DMGF on NF-κB signaling. (**A**) Western blot analysis for total cellular NF-κB p65 and IκB expression in LPS-stimulated splenocytes. LPS-stimulated splenocytes were treated with 0, 0.3, or 0.6 μg/mL DMGF for 10 h. (**B**) DMGF inhibited NF-κB transcriptional activity in HEK293 cells. Levels of NF-κB-GFP were calculated according to the fluorescent mean. The data of NF-κB transcriptional activity were normalized to the PMA treated group. *—*p* < 0.05, compared to the PMA group.

**Figure 5 biomedicines-09-01778-f005:**
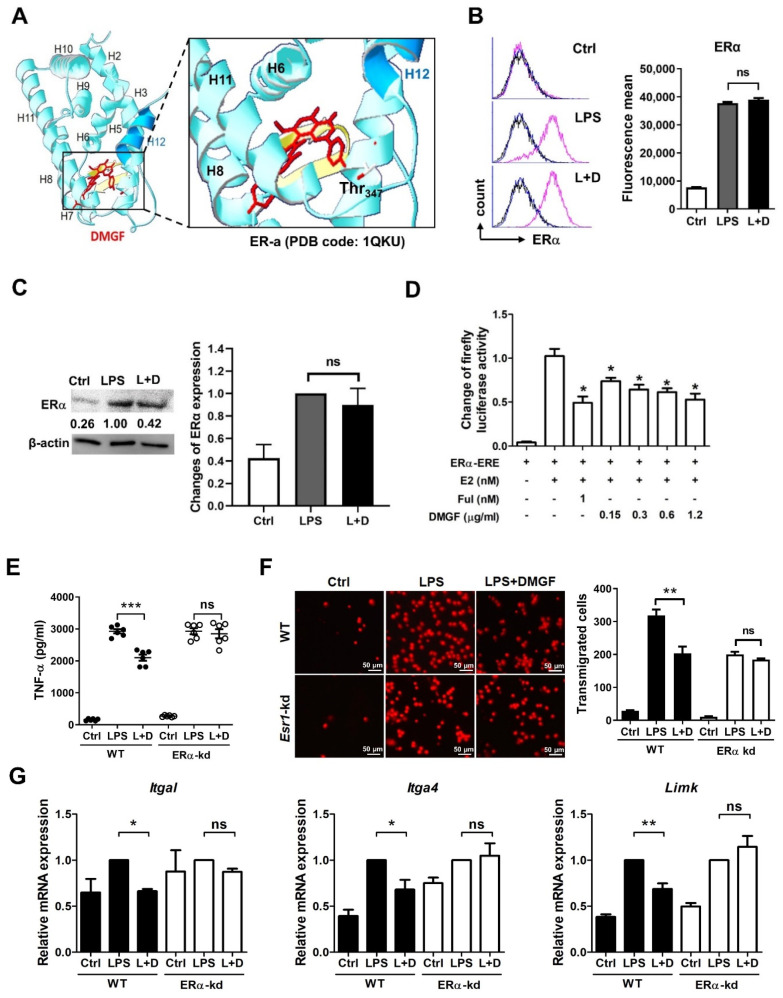
DMGF inhibited inflammation and cell migration through downregulating ERα transcriptional activity. (**A**) Docking of DMGF into the ligand-binding site of ERα (PBD code: 1QKU). The amino acid Thr347 that potentially interacted with DMGF is highlighted. (**B**) DMGF did not affect the expression of ERα on cell membrane in LPS-stimulated macrophages. The ERα expression in RAW264.7 cells was stained with anti-ERα antibody (pink) and control antibody (blue) and measured by flow cytometry. The black line indicates cells without staining. (**C**) DMGF did not affect total cellular ERα expression in LPS-stimulated macrophages. (**D**) DMGF inhibited ERα transcriptional activity in HEK293 cells. ERα-ERE Firefly luciferase activities were calculated with internal control of Renilla luciferase activities (n = 6). Estradiol (E2, 0.5 nM) was used to induce ERα transcription. Fulvestrant (Ful, 1 nM), an ER antagonist, was used as the positive control. The ERα transcriptional activity data were normalized to the E2-treated group. *—*p* < 0.05, compared to the E2 group. Inhibitory effects of DMGF on (**E**) TNF-α release, (**F**) cell transmigration, (**G**) and expression of *Itgal*, *Itga4* and *Limk* in LPS-stimulated RAW264.7 or LPS-stimulated RAW264.7-Esr1 knockdown cells. *—*p* < 0.05, **—*p* < 0.01, ***—*p* < 0.001, compared to the LPS group. ns, ns, no significant difference. The scale bar in the picture of transmigrated cells represents 50 μm.

**Figure 6 biomedicines-09-01778-f006:**
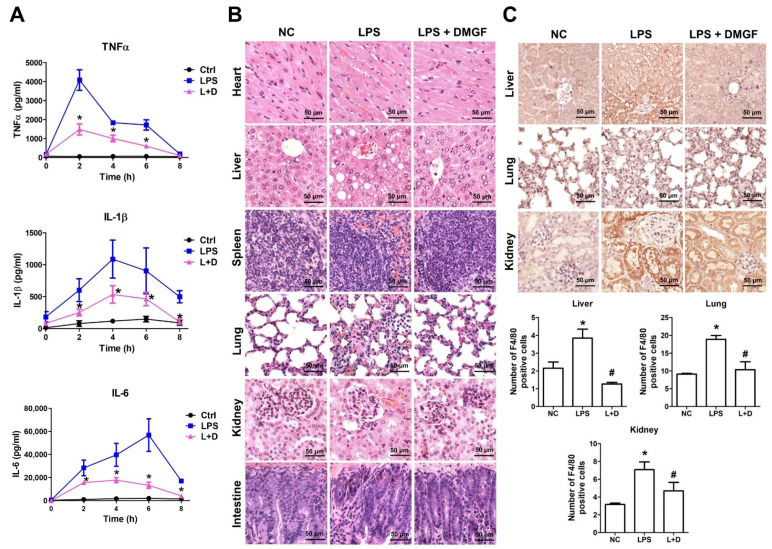
DMGF effectively reduced production of pro-inflammatory cytokines and organ damage in endotoxic mice. (**A**) DMGF reduced levels of TNFα, IL-1β, and IL-6 in sera. BALB/c mice received DMGF or vehicle (1% DMSO in PBS) by i.v. injection immediately after i.p. injection with LPS. Serum samples were collected and the levels of TNFα, IL-1β, and IL-6 were determined by ELISA. *—*p* < 0.05; compared to the LPS group. (**B**) Histopathological features of organs in different treatment groups. Organ sections were stained with H&E and imaged at 400× magnification. The scale bar in each picture represents 50 μm. (**C**) Infiltrating macrophage expression in the liver, lung, and kidney. Macrophages were stained with anti-mouse F4-80 antibody and imaged at 400× magnification. The average number of infiltrating macrophages was counted from ten random fields each slide. *—*p* < 0.05, compared to the NC group. #—*p* < 0.05, compared to the LPS group.

**Figure 7 biomedicines-09-01778-f007:**
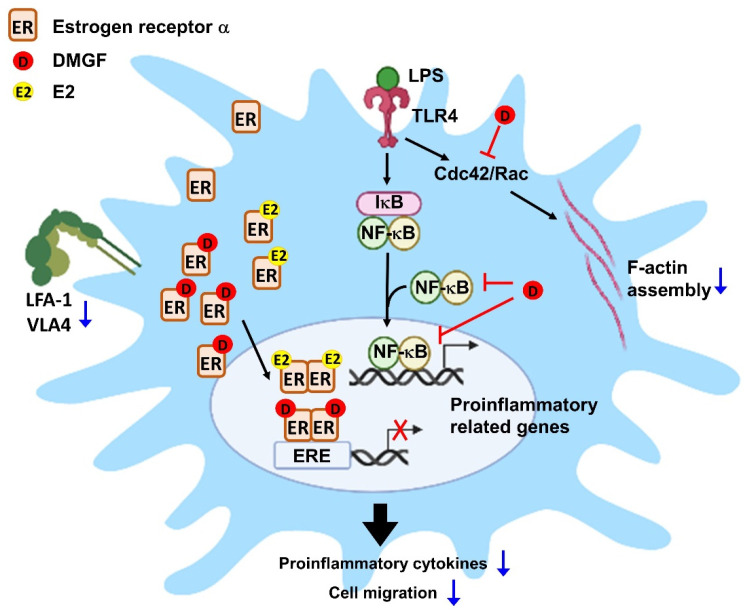
The schematic diagram shows the possible mechanism by which DMGF inhibits LPS-induced inflammation and cell migration. DMGF downregulated LPS-induced NF-κB expression and its transcriptional activity. Meanwhile, DMGF downregulated the transcriptional activity of ERα by interacting with ERα and then competitively abolishing E2-ERα binding on ERE. The two pathways downregulated by DMGF led to a decrease in the expression of pro-inflammatory-related genes, thereby effectively inhibiting inflammatory cytokines and cell migration in LPS-stimulated macrophages. ER—estrogen receptor α; D—DMGF, E2—estradiol; ERE—estrogen-response element; TLR4—Toll-like receptor 4. Blue arrows mean inhibition. The black arrows indicate the signal propagation of LPS induced inflammation. The blue arrows indicate the inhibitory effects after DMGF treatment.

**Table 1 biomedicines-09-01778-t001:** Effects of DMGF on cytokines and chemokines expression in LPS-stimulated splenocytes.

Cytokines and Chemokines	Control	LPS	LPS + DMGF
IL-1α	ND	8.6 ± 1.6	5.0 ± 0.6
IL-1β	0.24 ± 0.28	12.1 ± 0.7	8.2 ± 0.8 *
IL-2	0.60 ± 0.06	2.8 ± 0.6	1.9 ± 0.4
IL-3	0.09 ± 0.09	0.58 ± 0.03	0.50 ± 0.10
IL-4	0.08 ± 0.02	0.85 ± 0.18	0.63 ± 0.09
IL-5	0.49 ± 0.20	2.2 ± 0.4	1.8 ± 0.1
IL-6	1.1 ± 0.3	363.1 ± 50.2	259.9 ± 63.7 *
IL-9	ND	3.2 ± 0.2	2.8 ± 0.1
IL-10	0.56 ± 0.37	59.8 ± 15.7	47.8 ± 17.2
IL-12 (p40)	14.7 ± 6.8	108.1 ± 17.0	84.1 ± 17.5
IL-12 (p70)	3.9 ± 3.2	54.2 ± 3.8	40.6 ± 3.9
IL-13	ND	13.3 ± 1.6	8.3 ± 4.7
IL-17A	0.25 ± 0.19	1.5 ± 0.2	1.0 ± 0.4
Eotaxin	0.14 ± 0.12	3.5 ± 0.9	2.5 ± 0.5
G-CSF	ND	465.4 ± 96.9	427.9 ± 69.6
GM-CSF	ND	16.3 ± 0.7	10.6 ± 2.2
IFN-γ	0.76 ± 0.7	8.2 ± 1.5	4.6 ± 0.3
KC	0.30 ± 0.23	442.6 ± 56.2	323.1 ± 47.0 *
MCP-1	4.6 ± 9.2	131.3 ± 10.8	123.5 ± 10.4
MIP-1α	4.8 ± 1.0	1221.9 ± 76.0	652.1 ± 94.2 *
MIP-1β	46.2 ± 2.1	1397.1 ± 83.6	957.0 ± 25.1 *
RANTES	109.7 ± 17.4	2664.0 ± 405.7	2183.9 ± 447.8
TNF-α	13.5 ± 3.9	180.5± 37.9	89.9 ± 38.1 *

* All values are mean ± SD (n = 4). *—*p* < 0.05, compared to LPS group. ND, no detection.

## Data Availability

Not applicable.

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
