# Peer review of "7,7″-Dimethoxyagastisflavone Inhibits Proinflammatory Cytokine Release and Inflammatory Cell Recruitment through Modulating ERα Signaling"

_biomedicines, 2021, doi:10.3390/biomedicines9121778_

Round 1

Reviewer 1 Report

Dear Editor and Authors,

I kindly appreciate the effort that has been made by the authors in adressing the reviewer’s comments. This second version of the paper has been significantly improved. However, considering the potential applications and the relevance of the work, a few important points still need to be adressed by performing additional functional experiments and by making some changes in the text to improve its accuracy :

Concerning the functional experiments, you already performed lentiviral transduction knock-down experiments for ER-alpha and measured TNF-alpha production in LPS/DMSF stimulated cells, as suggested. You got pretty convincing results showing that, at least for this cytokine, DMGF effect was abolished in the absence of ER-alpha. However, a control for knock-down efficiency is still missing. Please, add it in the panel or as a supplemental figure and refer to it in the text. Moreover, since the effect of DMSF not only impacts cytokine production, but also cell migration, surface immune receptor expression and F-actin polymerization (as it appears in the working model), it would be crucial to perform additional ER-alpha knock-down in these 3 different experiments to clearly confirm the key mechanistic role that you claim for ER-alpha in limiting these different aspects of inflammation.

Concerning the changes in the text, please, address the following points :

  • The use of PMA instead of LPS in HEK cells should be also clearly justified in the text of the article to avoid confusion (the explanation for the reviewer was indeed pretty clear).
  • The pBIND-ER-alpha vector experiments show that the ER-ligand binding domain is important in DMGF sensing, but do not constitute evidence for direct binding of DMGF to the ER-ligand-binding domain per se. You would need to perform crystallography experiments to support such a strong statement. Please, be careful in the results section and the discussion and explicitely consider that, even if we have some in silico evidence, we can not exclude the participation of other cofactors in DMGF sensing. In this sense, also tone down the sentence “DMGF was able to interact with ligand binding domain”(line 23)
  • I kindly appreciate the addition of a working model. However, as it states, some points are confusing : In your scheme, DMGF seems to activate ER-alpha signaling, whereas your data show the opposite (Fig 5D for example). It also seems that DMGF inhibits LPS/TLR4 interaction. These concerns highlight a key question that has not been clearly discussed : How does ER-alpha/DMSF complex interfere with proinflammatory cytokine production? Please, briefly and clearly describe the possible mechanisms in the legend to guide the readers.
  • Concerning the title, the original one would be perfect if the additional functional experiments to prove the key role of ER-alpha are added. However, I do not understand why the authors proposed such an strange expression (“possesses the inhibition”), after the professional English editing.

Finally, there are still several typos and unaccurate terms (grammar/precision in describing the results) that have to be corrected :

  • Line 32: Aspergillus flavus -> italics
  • Line 66 : endothelium cells -> endothelial cells
  • Line 76 : endothelium -> endothelial
  • Line 77 : Integrins on monocytes -> Integrins expressed at the surface of monocytes
  • Line 97 : that contributed to suppressing -> thus reducing
  • Line 219 : Renilla was removed from the plasmid names. Please, check the consistency of the rest of the paragraph with the new names.
  • Line 275 : Non cytotoxic concentration of DMFG was used -> Non cytotoxic concentrations of DMFG were used
  • Line 282 : decrease -> decreases
  • Line 285 : molecules for adhesion -> adhesion molecules
  • Line 287 : by -> upon
  • Line 301 : show the actin fiber -> show actin fibers
  • Line 344 : inflammatory migration -> Migration of inflammatory cells
  • Line 368 : Expressions -> expression
  • Line 380 : improved -> limited
  • Line 383 : Suppressed -> reduced (tone down, it is far from being a 100 % effect if we considera ll the time points. There is still cytokine production between 2 and 6 hours.)
  • Line 387 : inhibited -> reduced
  • Line 388 : suppressed -> reduced
  • Line 411 : inhibited -> Inhibits
  • Figure 7 : assembling -> assembly
  • Line 433 : and “probably” before “through inhibiting ER-alpha transcription” , since you did not perform the functional experiments for cell adhesion molecules expression and F-actin polymerization by knocking-down ER-alpha.
  • Line 435 : anti-migration -> anti-migratory
  • Line 445 : bond -> bind
  • Line 451 : treatment of inflammation -> anti-inflammatory effect
  • Line 452 : are known to have -> is known to have (family, singular)
  • Line 459 : blockade -> block
  • Line 460 : inhibiting -> reducing
  • Line 461 : Camellia oleifera -> italics
  • Line 468 : suppression -> reduction
  • Line 471 : Suppress the Word “showed”
  • Line 487 : Suppress the Word “potent”. You already said potential before.
  • Figure S1 : in the supernatants were determined -> was (singular)
  • Figure S2 : ESR1 expression were determined -> was (singular)
  • Figure S3 : Total cellular NF-kB (p65) expression were determined -> was (singular) / the concentration in the supernatants in each group were determined -> was (singular)

Author Response

Dear Editor and Reviewers,

We appreciate all the valuable comments provided by the Editor and the Reviewers for helping us improve the readability of our paper. We have carefully examined all comments, suggestions, and requests and has conducted some modification as suggested.

  1. Concerning the functional experiments, you already performed lentiviral transduction knock-down experiments for ER-alpha and measured TNF-alpha production in LPS/DMSF stimulated cells, as suggested. You got pretty convincing results showing that, at least for this cytokine, DMGF effect was abolished in the absence of ER-alpha. However, a control for knock-down efficiency is still missing. Please, add it in the panel or as a supplemental figure and refer to it in the text. Moreover, since the effect of DMSF not only impacts cytokine production, but also cell migration, surface immune receptor expression and F-actin polymerization (as it appears in the working model), it would be crucial to perform additional ER-alpha knock-down in these 3 different experiments to clearly confirm the key mechanistic role that you claim for ER-alpha in limiting these different aspects of inflammation.

Response:

The Esr1 knockdown efficiency was shown in supplementary figure 3. The levels of Esr1 expression were normalized to the control siRNA. According to the reviewer’s comments, cell migration, integrin expression (Itgal and Itga4), and F-actin polymerization (Limk) were performed in ER-alpha knockdown cells. Results in Figure 5F and 5G showed that the inhibitory effects of DMGF on cell transmigration, integrin expression, and F-actin polymerization were reversed in Esr1 knockdown cells after DMGF treatment.

  1. The use of PMA instead of LPS in HEK cells should be also clearly justified in the text of the article to avoid confusion (the explanation for the reviewer was indeed pretty clear).

Response:

To avoid confusion with the cell model and activator we used for the NF-κB reporter assay. We added the description of the use of PMA instead of LPS in HEK cells in sub-section "2.10. NF-κB-GFP transcriptional reporter assay".

  1. The pBIND-ER-alpha vector experiments show that the ER-ligand binding domain is important in DMGF sensing, but do not constitute evidence for direct binding of DMGF to the ER-ligand-binding domain per se. You would need to perform crystallography experiments to support such a strong statement. Please, be careful in the results section and the discussion and explicitely consider that, even if we have some in silicoevidence, we cannot exclude the participation of other cofactors in DMGF sensing. In this sense, also tone down the sentence “DMGF was able to interact with ligand binding domain”(line 23).

Response:

We tone down the sentence “DMGF was able to interact with ligand-binding domain” as “DMGF might interact with ligand-binding domain” and show it in line 22.

  1. I kindly appreciate the addition of a working model. However, as it states, some points are confusing: In your scheme, DMGF seems to activate ER-alpha signaling, whereas your data show the opposite (Fig 5D for example). It also seems that DMGF inhibits LPS/TLR4 interaction. These concerns highlight a key question that has not been clearly discussed: How does ER-alpha/DMSF complex interfere with proinflammatory cytokine production? Please, briefly and clearly describe the possible mechanisms in the legend to guide the readers.

Response:

Thanks for the reviewer’s valuable comments. We modified the working model and shown in Figure 7. DMGF downregulated LPS-induced NF-κB expression and its transcriptional activity. Meanwhile, DMGF downregulated the transcriptional activity of ERa by interacting with ERa and then abolishing competitively E2-ERa binding on ERE. The two pathways down-regulated by DMGF lead to a decrease in the expression of pro-inflammatory-related genes, thereby effectively inhibiting inflammatory cytokines and cell migration in LPS-stimulated macrophages. This description has been added to the legend.  

  1. Concerning the title, the original one would be perfect if the additional functional experiments to prove the key role of ER-alpha are added. However, I do not understand why the authors proposed such a strange expression (“possesses the inhibition”), after the professional English editing.

Response:

We agree with the reviewer’s ideas. We still use the original title after the additional functional experiments that proved the key role of ER-alpha.

  1. Finally, there are still several typos and inaccurate terms (grammar/precision in describing the results) that have to be corrected:
  • Line 32: Aspergillus flavus -> italics
  • Line 66: endothelium cells -> endothelial cells
  • Line 76: endothelium -> endothelial
  • Line 77: Integrins on monocytes -> Integrins expressed at the surface of monocytes
  • Line 97: that contributed to suppressing -> thus reducing
  • Line 219: Renilla was removed from the plasmid names. Please, check the consistency of the rest of the paragraph with the new names.
  • Line 275: Non cytotoxic concentration of DMFG was used -> Non cytotoxic concentrations of DMFG were used
  • Line 282: decrease -> decreases
  • Line 285: molecules for adhesion -> adhesion molecules
  • Line 287: by -> upon
  • Line 301: show the actin fiber -> show actin fibers
  • Line 344: inflammatory migration -> Migration of inflammatory cells
  • Line 368: Expressions -> expression
  • Line 380: improved -> limited
  • Line 383: Suppressed -> reduced (tone down, it is far from being a 100 % effect if we considera ll the time points. There is still cytokine production between 2 and 6 hours.)
  • Line 387: inhibited -> reduced
  • Line 388: suppressed -> reduced
  • Line 411: inhibited -> Inhibits
  • Figure 7: assembling -> assembly
  • Line 433: and “probably” before “through inhibiting ER-alpha transcription” , since you did not perform the functional experiments for cell adhesion molecules expression and F-actin polymerization by knocking-down ER-alpha.
  • Line 435: anti-migration -> anti-migratory
  • Line 445: bond -> bind
  • Line 451: treatment of inflammation -> anti-inflammatory effect
  • Line 452: are known to have -> is known to have (family, singular)
  • Line 459: blockade -> block
  • Line 460 : inhibiting -> reducing
  • Line 461 : Camellia oleifera -> italics
  • Line 468 : suppression -> reduction
  • Line 471 : Suppress the Word “showed”
  • Line 487 : Suppress the Word “potent”. You already said potential before.
  • Figure S1 : in the supernatants were determined -> was (singular)
  • Figure S2 : ESR1 expression were determined -> was (singular)
  • Figure S3 : Total cellular NF-kB (p65) expression were determined -> was (singular) / the concentration in the supernatants in each group were determined -> was (singular)

Response:

Thanks for the reviewer’s friendly reminder. All the above typos and inaccurate terms (grammar/precision in describing the results) have been corrected in the revised manuscript and highlighted in blue.

Best regards

Yu-Ling Lin, Ph.D.

Assistant Research Fellow

Agricultural Biotechnology Research Center

Academia Sinica

128 Academia Road, Section 2, Nankang, Taipei 11529, Taiwan

Tel: 886-2-27872126

E-mail: lyring@gate.sinica.edu.tw

Reviewer 2 Report

Excellent work. I am very pleased with the positive efforts that authors have put through to improve the quality of the manuscript during the revision. Very well done.  The manuscript is very well written and conclusions are clearly delineated.

Author Response

Dear Editor and Reviewers,

We appreciate all the valuable comments provided by the Editor and the Reviewers for helping us improve the readability of our paper. 

Best regards

Yu-Ling Lin, Ph.D.

Assistant Research Fellow

Agricultural Biotechnology Research Center

Academia Sinica

128 Academia Road, Section 2, Nankang, Taipei 11529, Taiwan

Tel: 886-2-27872126

E-mail: lyring@gate.sinica.edu.tw

Round 2

Reviewer 1 Report

Dear authors,

Thanks for adressing all my comments. I point out that the efficiency of the siRNA for ER-alpha is low. However, the funcional results presented, although slight, seem significant. It will be crucial to improve this kind of depletion experiments in the future to maximize your effects and improve the significance and soundness of the results.

I thus accept this paper for publication.

Best regards

This manuscript is a resubmission of an earlier submission. The following is a list of the peer review reports and author responses from that submission.

Round 1

Reviewer 1 Report

In this manuscript, Wu et al. present the anti-inflammatory effect of DMGF, a biflavonoid isolated from the needles of Taxus x media var. Hicksii, in an inflammatory context induced by LPS both in vitro and in vivo. They show that DMGF is able to limit cytokine production, cell surface adhesion molecule expression and cell migration. Overall, the manuscript is well presented and adds to the field since it proposes a new clue to develop novel treatments against excessive inflammation. However, there are still several concerns that should be adressed before publication, especially regarding the cellular mechanisms that drive the anti-inflammatory effects :

Major concerns :

Although the cellular effects driven by DMGF are very clear (limitation of cytokine production, limitation of cell migration, limitation of surface adhesion molecules expression) some experiments performed to unveil the signalling mechanisms involved in the anti-inflammatory effects are not properly addressed :

  • Nf-kB and IkB downregulation are not clearly demonstrated : Figure 4A : How many times was this western blot performed ? Although the downregulation of Nf-kB in response to DMGF seems pretty clear, I don’t see any clear upregulation in IkB expression, as it is claimed by the authors… Please, repeat this experiment at least 3 times and quantify it properly.
  • LPS stimulation is lacking in some key experiments : Figures 4B et 5C : throughout the study, DMGF was used in an inflammatory context induced by LPS. However, in these figures, different agonists than LPS were used (PMA for Nf-kB induction and E2 for ER-alpha activation). What is the rationale for avoiding LPS stimulation ? This is inconsistent with the rest of your study. Please, include LPS stimulation in these two experiments to properly assess the effect of DMGF in Nf-kB and ER-alpha dependent signaling.
  • ER-alpha expression is not affected by DMGF in an LPS-induced inflammatory context, is it thus actually a key signalling actor ? : Figure 5B : I clearly see that DMGF did not affect the expression of ER-alpha in LPS-stimulated macrophages. Then, what would be the impact of mRNA expression reduction of ER-alpha (experiments still to be performed with LPS stimulation) ? Protein degradation could also be inhibited, thus leaving total ER-alpha levels unchanged… I suggest to invalidate ER-alpha (siRNA, for example) in cytokine production experiments in response to LPS and LPS+DMGF to verify its implication in the process. Since you only present in silico evidence for DMGF /ER-alpha interaction and experimental evidence for DMGF action on ER-alpha transcriptional activity independently of LPS stimulation, this experiment would at least prove functionally that ER-alpha is indeed involved in the effects that you claim (lowering cytokine production in an inflammatory LPS-dependent context).
  • A working model would be very valuable. I suggest the authors to add a working model at the end of the manuscript to recapitulate their findings and highlight theirs strenghts. The figure could include the cellular effects and the potential molecular mechanisms driven by DMGF.
  • The title of the study highlights the contribution of ER-alpha signaling in the anti-inflammatory effects of DMGF. However, this conclusion is only (very) partially supported by the data presented since most of the experiments on ER-alpha expression lack a proper inflammatory context induced by LPS (as it is the case for the rest of the experiments)

Minor points :

Although the paper is overall well written, it still has several grammatical mistakes, typos and unclear senteces that have to be corrected. Please, consider re-reading by a antive english speaker before resubmission. There are also some minor formatting issues in several figures.

  • There are greek letters (alpha, mu, etc…) missing throughout the text (especially in the units and the name of several key actors like ER-alpha, Nf-kB, etc.). Please, correct and read carefully before submitting it again.
  • Latin phrases like « in vitro », « in vivo » and species names should be written in italics.
  • Some sentences are not precise enough : line 14 (inhibition of cytokines : does it refer to « production », « release », « transcription » ?
  • The style of some sentences should be corrected to be more precise and also fit with proper english : line 251, it would be more accurate to say « to understand the anti-inflammatory role », « a non-cytotoxic concentration of DMGF was used ». Indeed, as it states, « anti-inflammation » is not accurate and « non-cytotoxic DMGF » implies that you use different compounds, some that are cytotoxic and some that are not… As well, « the concentration of different pro-inflammatory cytokines was measured » would be more accurate than « cytokines were detected », which just means if they were present or not…
  • Line 55 : « attract immune cell recruitment » -> « attract immune cells » or « increase immune cell recruitment »
  • Line 66 : « endothelium cells » -> « endothelial cells »
  • Line 112 : « Petri » instead of « petri ». it is a family name.
  • Line 237 : which microscope was used ?
  • Table 1 : Could you format it as a graph ? It would be easier for the readers to appreciate the anti-inflammatory effect of DMGF at a glance.
  • Line 249 : « verified » -> « assessed »
  • Line 265 : « anti-migration » -> « DMGF decrease macrophage migration »
  • Line 279/280 : You state « These results suggested that DMGF inhibition of macrophage migration is involved in F-actin polymerization » This sentence is confusing. You should say « these results suggested that the inhibition of macrophage migration in response to DMGF depends on a mechanisms relying on F-actin polymerization» As it states, it seems that DMGF is inhibited and that the phenomenon of migration itself is the basis for F-actin polymerization…
  • Line 281 : « filopodia-regulated proteins » and « lamellipodia-regulated proteins » are both incorrect. These proteins are not regulated by filopodia nor lamellipodia. It is the regulation of these proteins that drives the formation of this kind of structures…
  • Figure 2A : Was this experiment performed in a Boyden chamber ? Please, specify the technique used. In this case, you should refer to the cells as « transmigrated cells »
  • Figure 2C/D : The contrast/intesity of the images is not accurate. We barely don’t see the colocalization of both stainings… Please, adjust.
  • Line 302 : the title « Anti-inflammatory activity of DMGF through its inhibition…» should be « Anti-inflammatory activity of DMGF through the inhibition…». Moreover, the title implies that ER-alpha expression is implicated in the anti-inflammatory activity, whereas the experiments performed in figure 5C lack an inflammatory stimulus (LPS) and do not assess proinflammatory cytokine production…
  • Line 315 : « indicated/highlighted » instead of « visualized »
  • Figure 4B : Add B) in the figure description
  • Figure 5B : what are the conditions corresponding to the blue, black and pink curves ? which was the technique used ? Please, specify.
  • Line 371 : « clearly shows the potency ». Please, tone down « suggests the potential »
  • Figure 6A : Please, increase the size of the geometric forms or use a color code, we barely do not distinguish the curves.
  • Figure 6B/C : Please, add the scale. It seems that some pictures have different scales, especially in the panel C that corresponds to kidney.
  • Figure 6C : Remove the « s » from « numbers » in the graph
  • Line 361 : « anti-inflammation » should be replaced by « in the inhibition of inflammation »
  • Line 373 remove « and » between « cell and signalling »
  • Line 388 « promoter » instead of « promotor »
  • Conclusion : line 425 : You do not show direct experimental evidence that DMGF binds ER-alpha, so the conclusion that it is an antagonist is too strong. Please, tone down.
  • Conclusion : line 427 : DMGF does not « improve » tissue damage, it « limits » it.

Reviewer 2 Report

In this study, Wu et al examined the effects of DMGF on cytokine production and cell migration of inflammatory macrophages and found that DMGF inhibits inflammatory cascades in LPS-treated macrophages as shown by decreased pro-inflammatory cytokines and chemokines levels. In addition, DMGF was found to suppress macrophage movement through inhibition of adhesion molecules and destabilizing F actin polymerization. A docking of DMGF into the ligand binding site of ERα showed that DMGF interacts with ERα and also suppress the transcriptional activity along with downregulation of NF-kappa B. In an endotoxin mouse model, the authors further  showed the evidence that release of inflammatory mediators are suppressed upon treatment with DMGF along with reduced macrophage infiltration and damages in liver, lung and kidney. Overall, the authors have presented very interesting concept supported by extensive data. This study certainly holds a great interest to scientific community in general. The manuscript is well written and conclusions are carefully drawn. However, I have following comments which authors should address before considering the publication.

  • It is unclear how interaction between DMGF and ERα suppress ERα signaling? Do the authors have any explanation how binding to ERα can downregulate its transcriptional activity? This should be further discussed in detail.
  • What kind of antibody was used to detect ERα expression in figure 5B? The author should first provide evidence that the staining pattern is correct. Please provide gating information. Also mention the staining protocol in Method section. The authors are advised to check expression by western blot and normalize with loading control if possible before drawing any conclusion precisely. It is unclear how expression of ERα normalized in the flow cytometry data.
  • For fig 2B, and Table 1, it is not a good practice to use arithmetic mean or use a student t test given a sample size of 4.
  • Fulvestrant has been already proved selective estrogen receptor degrader. What advantage does DMGF offer over fulvestrant or selective ER modulators? The authors should discuss further on this to increase the interest and visibility of this work. Does Fulvetrant has any effect on NFKB expression? It would be interesting to see the effect of fulvetrant on NF-kappa B p65 as well at protein level.